# Structure–dynamics decoupling in soft-colloid suspensions

Adrián Arenas-Gullo [1,2], Joaquím Clara-Rahola[3], Phil N. Segré[4], José Ruiz-Franco [1,2] ✉ & Alberto Fernandez-Nieves [1,2,5]

The accepted paradigm in materials science is that the internal structure of a material determines its macroscopic properties. This connection is reflected in the particle dynamics, which are known to become slower at length scales comparable to the mean interparticle distance. This implies that the $q$-dependent relaxation time, with $q$ the magnitude of the scattering wave vector, correlates with the structure factor of the material. This is known as de Gennes narrowing in simple atomic liquids, and is a behavior also seen in colloidal suspensions, where the quantities at play are more easily accessible experimentally. We here find that this familiar correlation can breakdown for soft-colloid suspensions. In both experiments and simulations, we find that the $q$-dependent relaxation time of the suspension does not follow the structure factor, but that instead, it remains unchanged relative to the length scale at which it is measured. We justify this unusual behavior by alluding to single-particle elasticity and how this aspect allows additional relaxation pathways for the characteristic time of the suspension to remain unaffected by the suspension structure. Our findings challenge the prevailing wisdom that the structure of a material unequivocally determines its properties.

A key paradigm in materials science is that the structure of a material determines its properties[1]. This is evident, for example, in diamond and graphite, whereby the unique electron behavior and associated band gaps result from its particular atomic organization[2], in nanoscale magnetic skyrmions, whose properties result from the structural arrangement of the spins[3], and in disordered hyperuniform materials, which propagate waves in a singular manner[4,5]. This connection is also reflected in the dynamic slowing down associated to the presence of spatial correlations in dense atomic fluids, as first predicted by de Gennes[6]. The behavior, known since then as de Gennes narrowing, reinforces the crucial link between structure and dynamics that is fundamental in condensed matter physics. It was first experimentally observed in liquid argon[7], and later in confined fluids[8,9] and metallic liquids[10,11], and in a variety of soft materials, including polymer solutions[12,13], as well as protein[14] and colloidal suspensions[15].

Colloids are Brownian particles suspended in a solvent[16]. In dilute suspensions, they exhibit diffusive behavior with dynamics characterized by a diffusion coefficient $D_0$. Hard-sphere colloids are particularly interesting because of their simplicity: they are non-interacting particles, except for excluded volume. Nonetheless, they exhibit rich phase and non-equilibrium behavior with increasing particle volume fraction $\phi = nV$, with $n$ the number density of particles and $V$ their volume. Indeed, they manifest an entropy-driven crystallization process at sufficiently high $\phi$[17], but can also be supercooled beyond the freezing line to eventually experience a glass transition and become a glass at $\phi_g^{HS} \simeq 0.58$[18,19]. The approach to the glass is characterized by a dramatic slowing down of the dynamics and an associated increase, by many orders of magnitude, of the structural relaxation time $\tau_\alpha$[20,21]. In the process, however, the most prominent peak in the structure factor, $S(q)$, with $q$ the magnitude of the scattering wave vector, reflecting

[1]Department of Condensed Matter Physics, University of Barcelona, Barcelona, Spain. [2]Institute of Complex Systems (UBICS), University of Barcelona, Barcelona, Spain. [3]KHN Captial Consulting SL, Girona, Spain. [4]Emory Oxford College, Oxford, GA, USA. [5]ICREA-Institució Catalana de Recerca i Estudis Avançats, Barcelona, Spain. ✉e-mail: jmruizfranco@ub.edu

strong short-range positional correlations between particles, slightly but steadily increases with $\phi$.

This connection between structure and dynamics extends to all $q$, as reflected via the relation $\tau_0/\tau_\alpha(q) = D(q)/D_0 = 1/S(q)$, with $\tau_0 = 1/(q^2 D_0)$ the diffusive time scale, and $\tau_\alpha(q)$ and $D(q)$ the structural relaxation time and effective diffusion coefficient of the system at certain $q$, for a given $\phi$[22]. Noteworthily, the idea that the dynamics of a system follows its structure is the fundamental pillar of glass-transition theories like Mode-Coupling, which employs the structural information encrypted in $S(q)$ to identify the glassy kinetic transition as an arrest of the local dynamics of the particles[23]. Indeed, numerical solutions of the MCT equations for hard-spheres[24,25] and for supercooled molecular liquids[26] show a largest $\tau_\alpha(q)$ at the peak of $S(q)$.

Unlike hard spheres, soft colloidal particles such as microgels and star polymers can deform and even compress, as they have finite, often small, elastic moduli. This softness enables pushing the glass transition to higher $n$, and hence higher $\phi$[27]. However, the exact value of $\phi$ at which this happens is hard to know, as the particles are often fuzzy in their periphery, have an ill-defined surface, and can further shrink and interpenetrate in non-trivial ways at sufficiently high $n$[28,29]. We thus often use the so-called generalized volume fraction $\zeta = nV_0$, where $V_0$ is the volume of an undeformed particle $V_0 = \frac{4}{3}\pi R_h^3$, with $R_h$ the hydrodynamic radius measured in dilute conditions, to parameterize particle concentration; $\zeta$ is then to be taken as a proxy for $n$, informing on the suspension state and whether it is affected by softness. Note that while $\phi \leq 1$, $\zeta$ is unbounded and can take on values beyond 1 due to particle deformation, shrinking, and/or interpenetration. Experimental observations with microgels[30], charged hard spheres[31,32], core–shell colloids[33] and star-like micelles[34] have reported that $D(q)$ develops a narrow minimum at the $q$ where $S(q)$ is maximum, in agreement with the structure–dynamics coupling seen in hard-sphere suspensions and atomic systems. There is, however, experimental work with soft giant micelles that challenges this scenario[35] and that report on a $q$-independent relaxation time, despite the presence of a pronounced structural peak in the scattering signal. Remarkably, the underlying reasons for this surprising break-up of the structure–dynamics coupling are not understood, indicating that additional work is required in order to solve the puzzle and identify the relevant physics behind it.

Realize that this decoupling is different from the also decoupled structrure and dynamics of supercooled liquids on their approach to the glass, as epitomized by the huge slow-down of the dynamics with a comparatively nearly unchanged $S(q)$, all as $\phi$ increases. The structure-dynamics decoupling in giant-micelles is at constant $\phi$ and manifests in $q$[35].

In this paper, we employ aqueous thermoresponsive microgel suspensions and show that for low temperature, where particles are highly swollen and softer, the structure and the dynamics decouple. In contrast, for larger $T$, where particles are less swollen and stiffer, the structure and the dynamics couple a la de Gennes narrowing, all at constant $\zeta$. We perform numerical simulations using a Multi−Hertzian (MH) interaction potential that qualitatively capture the observed phenomenology, demonstrating that single-particle elasticity plays a major role in the observed coupling and decoupling behavior.

## Results

### Microgel particles

Our microgels are comprised of poly-N-isopropylacrylamide (pNIPAM), cross-linked with poly-ethylene glycol diacrylate (pEGd); see Section "Microgels" in "Methods" and ref. 36 for synthesis details. The resultant particles are temperature sensitive with a polydispersity of ~10%[36]. They are swollen and have a hydrodynamic radius of $R_h \simeq 260$ nm at $T = 10\,°C$, as measured by dynamic light scattering (DLS), and progressively deswell down to $R_h \simeq 80$ nm at $T = 37\,°C$, as also measured by DLS. The radius of gyration, as measured by static light scattering (SLS), also decreases from $R_g \simeq 150$ nm at $T = 10\,°C$ to $R_g \simeq 60$ nm at $T = 37\,°C$. However, in the $T$-range experimentally accessed, $R_h$ and $R_g$ are not proportional to each other, indicating that the internal structure of the microgels changes with increasing $T$. This can be best illustrated through the ratio $R_g/R_h$, which informs about the mass distribution within the particles[37]. For homogeneous spheres, for instance, $R_g/R_h = \sqrt{3/5}$[38], while for many soft spheres, most commonly exhibiting a core–shell structure, $R_g/R_h \sim 0.61$[39]. Our microgels have $R_g/R_h \sim 0.57$ below $T = 17.5\,°C$, and $R_g/R_h \sim 0.67$ for $17.5\,°C < T < 32\,°C$, as shown in Fig. 1a. Above $T \simeq 32\,°C$, $R_g/R_h \sim 0.71$, as also shown in Fig. 1a. The structure of the microgels is thus markedly open at small $T$, with most mass concentrated near the particle center, consistent with

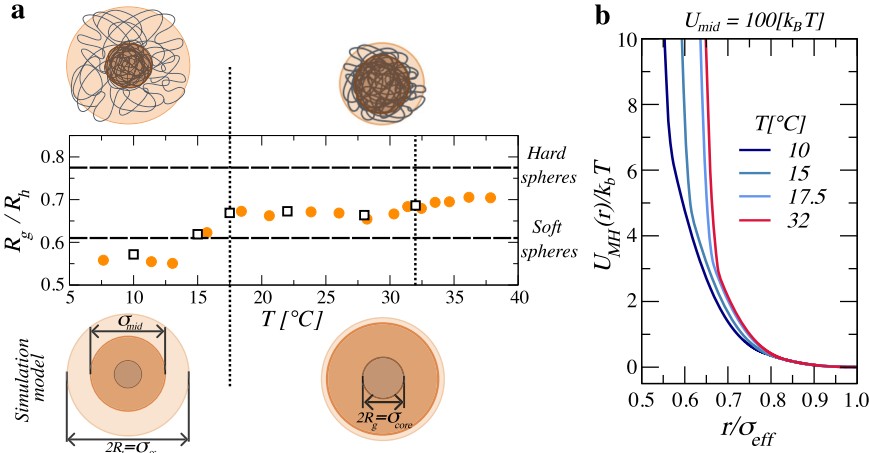

**Fig. 1 | Experimental system and simulation model. a** Ratio between the radius of gyration $R_g$ and the hydrodynamic radius $R_h$ of the microgels as a function of the temperature $T$. The value of this ratio is a measure of particle softness, as it reflects how the mass distributes within the microgels. Vertical lines highlight the transition temperatures between values we associated to different particle morphologies. The horizontal lines emphasize the characteristic values for soft and hard spheres. The sketches above the plot are meant to represent the internal structure suggested by the $R_g/R_h$ ratio, and prior SANS experiments[36]. The squares in the plot correspond to the temperatures at which we will measure both structure and dynamic properties at large $\zeta$. The schematics below the plot correspond to the simulated multi-layered particle structure, with highlighted geometrical features used to model softness together with the interaction strength associated to the various layers. In our model, the particle size $\sigma_{eff}$ remains unchanged, while the intermediate shell size $\sigma_{mid}$ and the core dimension $\sigma_{core}$ are both tuned to match the experimental $R_g/R_h$ values with $T$. **b** Interaction potential for $U_{mid} = 100\,k_BT$ and $U_{corona}=20\,k_BT$, and several representative temperatures. The value of $\sigma_{core}$ and $\sigma_{mid}$ for each T are given in Section 4 of the Supplementary Information.

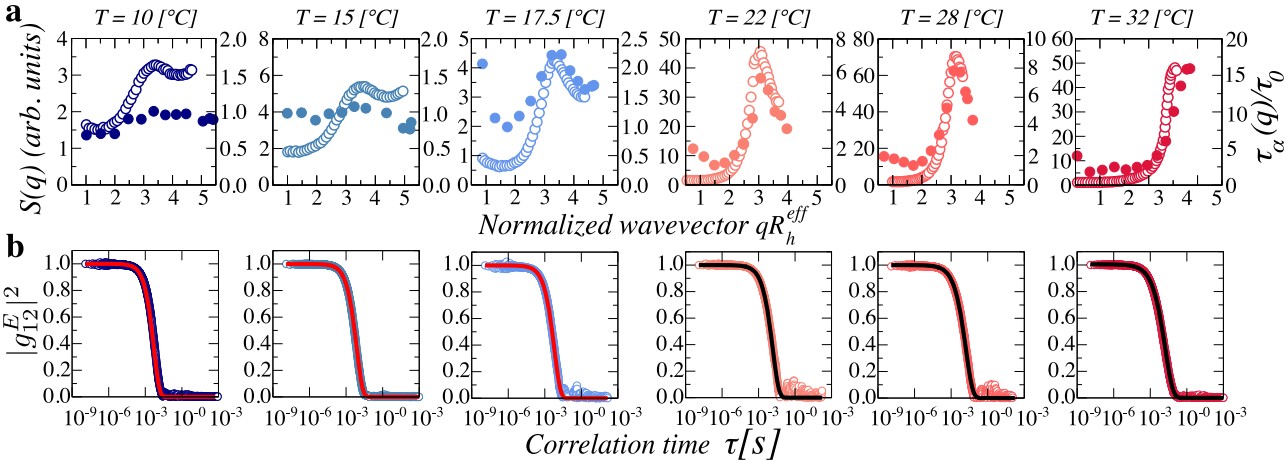

**Fig. 2 | Experimental results at $\zeta = 1.5$. a** Measurable structure factor $S(q)$ (empty symbols), and normalized structural relaxation time $\tau_\alpha(q)/\tau_0$ (filled symbols). **b** The square of the field correlation function $|g_{12}^E|^2$ as a function of $\tau$ measured at the peak of $S(q)$. The solid lines are the fits to a stretched exponential function. Up/down plots correspond to the indicated temperature.

the star-like form factor found using small angle neutron scattering[36,40]. It then evolves to that of typical soft spheres at intermediate temperatures, also consistent with form factor measurements[36], and approaches the structure of hard spheres at the highest $T$ we have accessed. The transition from soft to close-to-hard sphere behavior happens at $T_c \approx 32\,°C$, corresponding to the lower critical solution temperature of pNIPAM[41,42]. The less usual transition from a close-to-star-polymer to a soft sphere structure at lower temperatures has been attributed to the presence of pEGd[36]. At $T \approx 17\,°C$ pEGd segregates from pNIPAM, inducing internal microphase separation and heterogeneities within the microgel network[43]. This segregation causes the shell of the soft sphere to further stretch out at low $T$, bringing about a form factor that is well described using star-polymer models[36], and consequently modifying the particle softness. The morphological change of the particle with $T$ is schematically sketched on top of Fig. 1a.

### Dense suspensions

We then concentrate the suspension, fix $\zeta = 1.5$, and determine the structure factor at various temperatures across the swelling behavior; these $T$ are indicated with squares in Fig. 1a. Note that since the particle size decreases with increasing $T$, we increase $n$ at each $T$ to maintain a constant $\zeta$. We estimate $n$ by performing viscometry experiments[44]. We employ a multi-angle spectrometer (LS Instruments) using the so-called 3D DLS scheme that allows extracting single-scattering from samples exhibiting multiple-scattering[40,45,46], which often include relatively dense suspensions of swollen microgels. For a given $T$, we obtain at each $q$ the mean single-scattered intensity $\langle I^{(1)}(q)\rangle$, as discussed in Section "3D cross correlation light scattering" in "Methods", and use the corresponding dilute-suspension value, which effectively provides an estimate of the particle form factor $P(q)$, together with the particle number densities at $\zeta = 1.5$ and in the dilute regime, $n$ and $n_{dil}$, respectively, to obtain the structure factor $S(q) = \langle I^{(1)}(q)\rangle n_{dil}/[P(q)\,n]$. Note that we are obviating prefactors related to, for example, the scattering contrast, which may be different in the dilute and the dense regimes, where the particles may be forced to shrink. In addition, polydispersity at high $\zeta$ may differ from that in dilute conditions, and even at different $T$, due to changes in the particle morphology. This last aspect could also imply changes in $P(q)$ at $\zeta = 1.5$ relative to the dilute case that are hard to take into account, and that could affect our estimated $S(q)$. As a result, $S(q)$ should be taken as a reasonable estimate of the measurable structure factor. We then plot $S(q)$ versus $qR_h^{\text{eff}}$, where $R_h^{\text{eff}} = R_h/1.5^{1/3}$ accounts for the slight shrinking that must have happened at $\zeta = 1.5$ relative to the situation at $\zeta = 1$. We find that $S(q)$

exhibits a peak at a value of $q^* R_h^{\text{eff}} \approx \pi$, as shown in Fig. 2a and in Section 1 of the Supplementary Information, corresponding to the characteristic nearest neighbor peak and clearly reflecting the presence of spatial correlations between the particles. Note that this is true at all temperatures and thus irrespective of the swelling state and internal structure of the microgels. The less pronounced peak at $T < 17.5\,°C$ could relate to changes in $P(q)$ at $\zeta = 1.5$ relative to the dilute case not accounted for in our normalization of $\langle I^{(1)}(q)\rangle$, as mentioned above. The measurable structure factor, nevertheless, shows a clear increase from low $q$ to the peak, which additionally follows the expected trend when plotted against the particle size $R_h^{\text{eff}}$ in Section 1 of the Supplementary Information.

We simultaneously determine the normalized intensity correlation function at each $q$ and every $T$, and use the Siegert relation to obtain the square of the field correlation function, $|g_{12}^E|^2$, which is the square of the dynamic structure factor, often also noted as $F_c(q, \tau)$; see Section "3D cross-correlation light scattering" in "Methods". Examples of the result at fixed $q$ and all temperatures are shown in Fig. 2b. They all decay to zero for large correlation time $\tau$, indicating the ergodic character of the suspensions, even at the high $n$ required to have $\zeta = 1.5$. This behavior reflects the liquid-like character of all samples. Typically, in supercooled liquids and glasses, $|g_{12}^E|^2$ exhibits two decays[20,23,47]. The first decay, present at short times, is known as the $\beta$-relaxation mechanism and is associated with the microscopic motion of particles within their instantaneous nearest-neighbor cages. The second decay, emerging at longer times, is attributed to the $\alpha$-relaxation mechanism, corresponding to the structural relaxation of the system. In our case, we observe that $|g_{12}^E|^2$ exhibits a single decay, described by a stretched exponential, as shown by the corresponding fits in Fig. 2b. As a result, $|g_{12}^E|^2 = \exp[-(\tau/\tau_\alpha)^p]$, where $p$, referred to as the stretching exponent, and the structural relaxation time, $\tau_\alpha$, are obtained from the fits; see Section 2 of the Supplementary Information. We do this at all $q$ and find that while for $T > 17.5\,°C$, $\tau_\alpha(q)/\tau_0$ qualitatively follows $S(q)$, for $T < 17.5\,°C$, $\tau_\alpha(q)/\tau_0$ is essentially constant and independent of $q$; this is shown using filled circles in Fig. 2a. Note that $\tau_0 \sim q^{-2}$, even if we do not explicitly write the $q$ dependence. It is worth pointing out that, in order to account for particle shrinking, we actually use the diffusive timescale $\tau_0(R_h^{\text{eff}}) = 1.5^{-1/3}\tau_0(R_h)$. Additionally, while $p \approx 0.9$ for $T < 17.5\,°C$, corresponding to diffusive behavior, which is characterized by $p = 1$, for $T > 17.5\,°C$, $p \sim 0.75$, clearly indicating the subdiffusive character of the system, see Section 1 of the Supplementary Information. For $\zeta = 3.0$, $|g_{12}^E|^2$ exhibits a two decay behavior depending on $T$, indicating that caging effects become relevant at this higher $\zeta$. In this case, we focus on the final decay, which we fit to a stretched exponential to find

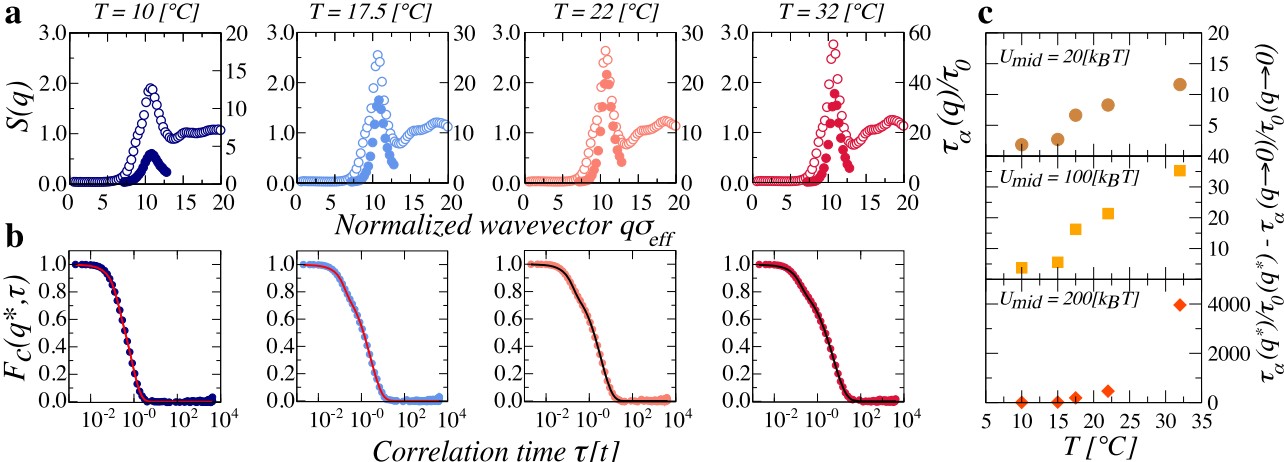

**Fig. 3 | Simulation results for $U_{mid} = 100\,k_BT$ at $\zeta = 1.8$. a** Structure factor $S(q)$ (empty symbols), and normalized structural relaxation time $\tau_\alpha(q)/\tau_0$ (full symbols). **b** Dynamic structure factor $F_c(q^*, \tau)$. Solid lines are fits to either a single stretched exponential or a sum of stretched exponentials. **c** Relative height of the structural relaxation peak, $\Delta\tau_\alpha = \tau_\alpha(q^*)/\tau_0(q^*) - \tau_\alpha(q \to 0)/\tau_0(q \to 0)$, as a function of $T$ for our softest particles with $U_{mid} = 20\,k_BT$ (top panel), for particles with intermediate stiffness with $U_{mid} = 100\,k_BT$ (middle panel), and for our stiffest particles with $U_{mid} = 200\,k_BT$ (lower panel).

$p$ and $\tau_\alpha(q)$; the qualitative behavior of these quantities and the coupling or decoupling of the structure and the dynamics at high and low $T$ is qualitatively similar to that found at $\zeta = 1.5$; see Section 3 of Supplementary Information.

The quantitative differences between $S(q)$ and $\tau_\alpha(q)$ seen in Fig. 2a at $T > 17.5\,°C$ have often been addressed by alluding to hydrodynamics. While for atomic systems the proportionality between these two functions is accurate, in suspensions, due to the presence of the solvent, the relation is mediated by another function, $H(q)$, that accounts for the role of hydrodynamic interactions[48]. Since microgels can be peripherically permeable to the solvent in amounts that depend on their swelling state[49], and hydrodynamic interactions between particles depend on this permeability effects[50], it may be that these effects could play a role in our comparison between $S(q)$ and $\tau_\alpha(q)$. However, these permeability effects are known to affect the extent of quantitative coupling between these quantities through changes in $H(q)$, but are unable to cause the dramatic decoupling we observe experimentally at low $T$[50]. Such a decoupling was previously observed in suspensions comprised of giant micelles[35]. There, it was argued that the polymer corona of the micelles could potentially facilitate their translational and rotational motion, aiding in the observed structure-dynamics decoupling and delaying the glass transition, even in the presence of pronounced local correlations as reflected by the presence of a clear peak in $S(q)$. Even if the underlying origin of the decoupling was unclear, the rotational aspect alluded to suggested that the internal degrees of freedom of the particles could play a role in the phenomenology. We then hypothesize that the relevant internal degrees of freedom can ultimately be thought of in terms of elastic properties, and that as a result, the single-particle elasticity plays a role in the decoupling of the structure and the dynamics. At low $T$, the microgels are softest and the large capability for deforming provides additional mechanisms for the structural relaxation of the suspension, which is nevertheless characterized by local positional correlations.

## Exploring the role of single-particle elasticity

To test this hypothesis, we perform coarse-grained numerical simulations considering spherical particles interacting through a MH potential[51]. This approach enables modeling our microgels as a set of two concentric spherical shells around a dense core. The core accounts for the highly cross-linked central region of most microgels[52,53], while the two shells around it allow accounting for the soft character of the particles and the segregation of pEGd from pNIPAM associated to the

change in $R_g/R_h$ at $T$ below $17.5\,°C$. To further connect with the experiment, we assume a constant overall particle size, $\sigma_{eff}$, and modify the size of the dense core, $\sigma_{core}$, to match the experimental values of $R_g/R_h$ at different $T$. The size of the intermediate shell is taken to be $\sigma_{mid} = 0.5(\sigma_{core} + \sigma_{eff})$; see Section "Simulation details" in "Methods". Considering this single-particle model, the pair potential we use in the simulations is:

$$U_{MH}(r) = U_{corona}[1 - r/\sigma_{eff}]^{5/2}\Theta(\sigma_{eff} - r)$$
$$+ U_{mid}[1 - r/\sigma_{mid}]^{5/2}\Theta(\sigma_{mid} - r) + U_{core}[1 - r/\sigma_{core}]^{5/2}\Theta(\sigma_{core} - r),$$

$$(1)$$

where the first, second and third terms account, respectively, for the overlap of the outer shell, the intermediate shell and the particle core, each with associated strengths $U_{corona}$, $U_{mid}$, and $U_{core}$, and $\Theta(\sigma - r)$ is the Heaviside function guaranteeing the interaction is only at play in the presence of the required overlap between particles. The effective single-particle elastic properties can then be tuned by modifying the extent of the different shells, $\sigma_{core}$ and $\sigma_{mid}$, see Section 4 of the Supplementary Information, and the interaction strength associated to the overlap of the shells, $U_{corona}$ and $U_{mid}$. We fix $U_{core} = 10^4\,k_BT$, with $k_B$ Boltzmann's constant, and always consider that $U_{corona} \le U_{mid} \ll U_{core}$. If we then choose $U_{corona} = 20\,k_BT$ and $U_{mid} = 100\,k_BT$, and vary the extent of the shells, the resultant pair potentials corresponding to the experimental values of $T$ are progressively softer as $T$ decreases, as shown in Fig. 1b; the corresponding particle morphologies are schematically shown below Fig. 1a. Additional simulation details are available in Section "Simulation details" of the "Methods" section.

We then fix $\zeta = 1.8$ and compute the suspension structure factor. We find that irrespective of the sizes associated to the core and shells of the particles, which are meant to capture the experimental $R_g/R_h$ changes with $T$, there is always a pronounced peak, as shown with open circles in Fig. 3a and highlighting the presence of strong spatial correlations between the particles. We also determine the dynamic structure factor at different $q$. These always decay to zero, as shown in Fig. 3b, emphasizing the liquid-like character of the suspensions. The long-time decorrelation of the dynamics is well captured with a stretched exponential decay that allows obtaining the structural relaxation time. Its $q$-dependence exhibits a peak at the $q$ of the structure factor peak but with a height that greatly depends on softness and hence on the $T$ of the experiment with $R_g/R_h$ values coinciding with those in our simulations; see Fig. 3a. To more clearly emphasize this fact, we plot

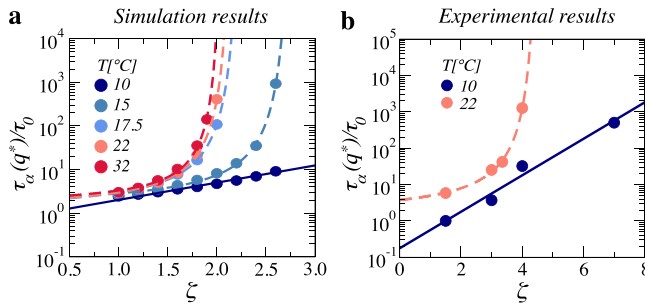

**Fig. 4 | ζ-dependence of the structural relaxation time. a, b** Relative relaxation time obtained **a** in simulation for $U_{mid} = 100\,k_BT$ and **b** in experiment at $T = 10\,°C$, where structure and dynamics decouple, and $T = 22\,°C$, where structure and dynamics are coupled. In both plots, solid lines represent fits to an Arrhenius exponential model, $\tau_\alpha(q^*)/\tau_0 \sim \exp[A\zeta]$, or a Vogel–Fulcher–Tamman law, $\tau_\alpha(q^*)/\tau_0 \sim \exp[A\zeta/(\zeta_0 - \zeta)]$. The fitting parameters are, respectively, $A$, or $A$ and $\zeta_0$.

the relative height, $\Delta\tau_\alpha = \tau_\alpha(q^*)/\tau_0(q^*) - \tau_\alpha(q \to 0)/\tau_0(q \to 0)$, with $q^*$ the magnitude of the scattering wave vector at the peak of $S(q)$, versus $T$, and observe that indeed $\Delta\tau_\alpha$ decreases with decreasing temperature, as shown in Fig. 3c. This indicates that the degree of coupling between the structure and the dynamics clearly depends on single-particle softness, confirming our hypothesis and highlighting the plausible role of this aspect in the experiments. Decreasing the interaction strength associated to the intermediate shell so that $U_{mid} = U_{corona} = 20\,k_BT$ results in a smaller increase of $\Delta\tau_\alpha$ with increasing $T$, indicating a less significant structure-dynamic coupling at high $T$; see Fig. 3c. On the contrary, considering $U_{mid} = 200\,k_BT$, results in a larger increase in $\Delta\tau_\alpha$ with increasing $T$, indicating a more pronounced structure-dynamic coupling at high $T$; see Fig. 3c. Additional numerical results for these two $U_{mid}$ values are shown in Section 5 of the Supplementary Information. Thus, our simulation results clearly show that single-particle elasticity can significantly affect the coupling between structure and dynamics. However, unlike the experimental observations, a complete erasure of the peak in $\tau_\alpha$ is not observed in simulations. In our model, particle deformation is captured through particle overlaps, which impose an energy barrier that must be overcome to allow for structural relaxation. Experimentally, our pNIPAM-pEGd particles can deform to accommodate crowding without any meaningful overlaps, suppressing these energy barriers. As a result, the peak in $\tau_\alpha(q)$ observed in simulations at low $T$ is absent in experiments, highlighting the distinct way particle softness manifests in the two approaches.

**Approach to the glass**

Prior work highlighted that particle softness can also influence the increase in the structural relaxation time $\tau_\alpha(q^*)$, with $\zeta$, as the suspension approaches the glass. While suspensions comprised of softer particles manifested an exponential increase of the structural relaxation time with $\zeta$, stiffer particles showed a more abrupt increase, consistent with a Vogel–Fulcher–Tamman (VFT) law[54]. To check whether the structure-dynamics coupling correlates with the functional form with which $\tau_\alpha(q^*)$ increases with $\zeta$, we perform additional simulations to quantify this dependence. We find that at high $T$, the behavior is consistent with a VFT law, while at the lowest $T$, it is well described by a single exponential, as shown in Fig. 4a; see also Section 6 of the Supplementary Information. This suggest a correlation that we can then test in our experiments. We choose $T = 10\,°C$, where structure and dynamics are decoupled, and $T = 22\,°C$, where structure and dynamics are coupled, and determine the $\zeta$-dependence of $\tau_\alpha(q^*)$. We find that the behavior is consistent with a VFT law at $T = 22\,°C$ and with a single exponential growth at $T = 10\,°C$, as shown in Fig. 4b, with corresponding details in Section 6 of the Supplementary Information;

this qualitatively agrees with the simulations, further confirming that there is a correlation between this functional relation and the structure-dynamics coupling.

Our findings could be taken further to suggest that the structure-dynamics coupling correlates with the fragility of the glass. For molecular glass-formers, the so-called strong glasses exhibit an exponential increase of the relaxation time with inverse temperature, while the so-called fragile glass-formers exhibit a behavior consistent with a VFT law. The trends in Fig. 4 could indicate a transition from a fragile behavior at high $T$ to a strong behavior at low $T$, consistent with prior work[54]. This would then indicate that strong glass-formers exhibit structure-dynamics decoupling, while fragile glass-formers exhibit structure–dynamics coupling. This is, however, not at all obvious. A main issue pertains to whether fragility is well, or can even be, defined using $\zeta$ as the independent variable. Strictly speaking, it is the actual volume fraction $\phi$, and not $\zeta$, that is the truly important state function[55]. A simple phenomenological model explicitly accounting for single-particle compressibility highlighted that if $\phi$ instead of $\zeta$ is used to define fragility, the apparently strong behavior completely disappears, implying that all approaches to the glass correspond to a fragile behavior[19]. This was also discussed in independent experiments with microgel and charged hard-sphere suspensions[56]. The origin of the sole fragile approach can be easily understood after realizing that while $\zeta$ increases linearly with $n$, $\phi$ must do so non-linearly, due to particle shrinking; the result is that the simple exponential growth of the structural relaxation time with $\zeta$ becomes a much more abrupt increase when plotted against $\phi$. Therefore, the strong approach to the glass in soft-particle suspensions could ultimately be an apparent effect, rendering the initially suggested correlation between the structure–dynamics coupling and fragility unwarranted.

We end by addressing this issue in our simulations. Although the numerical model we have chosen does not explicitly consider particle deformations, we can try to indirectly introduce this aspect by defining a non-overlap volume that would experimentally correspond to the volume of the deformed particles. We do this by computing the Voronoi volume $V_v$ for each particle; see Section 7 of Supplementary Information. Using $V_v$, we can then define an effective packing fraction $\phi_v$. When plotting $\tau_\alpha(q^*)$ versus $\phi_v$, we find that all $\tau_\alpha(q^*)$–$\zeta$ curves collapse onto a single master curve, with a behavior that would indicate fragile behavior, consistent with ref. 56.

## Discussion

In supercooled liquids and glasses, the onset of slow dynamics is governed by the increase in the structural relaxation time $\tau_\alpha$ as the volume fraction $\phi$ increases, while the static structure factor $S(q)$ remains essentially unchanged[57,58]. This scenario reflects a structure–dynamics decoupling, albeit in a sense that differs from the one associated with de Gennes narrowing. The latter occurs at constant $\phi$ and reflects how the dynamics slows down at length scales comparable to the interparticle separation. Hence, $\tau_\alpha$ increases around the wave vector $q$ at which $S(q)$ is peaked[59]. While the coupling between structure and dynamics holds for many soft materials, ranging from hard-spheres[24,25] to emulsion and polymeric systems[12,60], we demonstrate here that particle softness can introduce additional relaxation mechanisms to allow the system to relax on shorter time scales. As a result, even in regimes where structural features are prominent, these do not alone govern the dynamic properties, and hence, $\tau_\alpha$ can become $q$-independent.

Combining experiments on pNIPAM-pEGd microgels with numerical simulations at high generalized volume fractions $\zeta$ we illustrate this fact. Experimentally, pNIPAM-pEGd microgels exhibit a softness even below the typical value reported for core–shell particles at low temperatures. This softness allows the microgels to undergo structural relaxation that is decoupled from structural correlations; $\tau_\alpha$ is then independent of $q$. As the temperature increases,

the microgels become stiffer due to a transition associated with changes in their internal architecture. This change in elasticity results in the dynamics becoming dependent on static properties, and $\tau_\alpha$ qualitatively follows $S(q)$. This behavior is further supported by numerical simulations, which show that precise tuning of single-particle softness enables control over the extent to which static correlations guide dynamical properties. These findings demonstrate that single-particle softness can decouple structure from dynamics, challenging the traditional view that structural correlations alone dictate relaxation behavior. In this direction, we note that MCT relies on $S(q)$ to predict the wave vector-dependent dynamics and the growth of $\tau_\alpha$ as the system approaches the glass[23,61]. However, in the case of soft particles, like the pNIPAM-pEGd microgels studied here, additional relaxation pathways arise from the internal degrees of freedom associated with particle deformability and softness, enabling dissipative mechanisms that are not captured by $S(q)$ alone. As a result, the assumption that structural correlations fully determine the dynamics breaks down. This suggests that MCT, in its standard form, may not account for the relaxation mechanisms present in ultrasoft systems, and that modifications to the theory may be required to incorporate the influence of particle softness.

Although single-particle softness can introduce relaxation mechanisms to decouple structure and dynamics, additionally resulting in distinct functional dependences of the relation between the structural relaxation time and $\zeta$, it does not seem to enable changing the fragile character of the suspension approach to the glass. Note, however, there is work arguing differently[62]. It is likely that the way softness is defined and how it affects the suspension volume fraction, which is in no way trivial when shrinking and deformation are at play, what might lie behind this still unresolved issue, demanding further work on the subject. The structure-dynamics decoupling reported here is nevertheless a significant finding, as it may allow using single-particle properties to design materials that would escape the material science paradigm that structure determines properties. Experiments with ultra-low crosslinked microgels, which are characterized by being very soft particles[63,64], as well as with cellular systems, such as eukaryotic cells[65], which are soft and out-of-equilibrium, might be of interest to further inquire on possible ways to achieve structure–dynamics decoupling.

## Methods

### Microgels

Unless otherwise stated, all chemicals were acquired from Sigma-Aldrich. N-Isopropylacrylamide (NIPAM) is recrystalized from hexane (J. T. Baker). Poly(ethylene glycol) diacrylate (pEGd, $M_W = 700$ g/mol) is used as a cross-linker instead of the more widely used N,N'-methylenebisacrylamide (BIS). The surfactant used during the synthesis is sodium dodecyl sulfate (SDS), and ammonium persulfate (APS) is the initiator of the polymerization reaction. Finally, the water used for the synthesis was distilled and deionized up to a resistivity of 18.2 MΩ cm (Barnstead E-Pure).

We obtain the pNIPAM-pEGd particles via aqueous free radical precipitation polymerization[36]. We initially prepare a solution comprised of 98% (7.76 g) NIPAM and 2% (1.03 mL) pEGd, with an overall monomer concentration of 70 mM, in a volume of 1 L. This monomer solution is then mixed with 1 mM (0.2884 g) of SDS inside a 2 L three-neck round bottom flask, where it is stirred and purged with $N_2$ at a temperature of 70 °C for an hour. We then add 0.2282 g of APS dissolved in 1mL of deionized water to initiate the polymerization process. The temperature is kept at 70 °C and there is always $N_2$ flowing above the monomer solution. The reaction is allowed to proceed for 12 h. We then dyalize the product against ultrapure water, freeze-dry it and resuspend it in $H_2O$. In the last step, we perform three cycles where we raise the temperature to 42 °C, in order to deswell the particles and ensure Brownian dynamics, and lower it to the temperature at which

we will perform the experiments. This is done at the desired $\zeta$. We finally emphasize that the presence of pEGd reduces the effect of the characteristic hydrophobic interaction of pNIPAM above the LCST of NIPAM[66,67]; this is attributed to surface-segregation of pEGd, as previously observed using $^1$H NMR[66,67].

### 3D cross-correlation light scattering

Light scattering techniques are widely used in colloid science. Measuring concentrated systems has added difficulties, often related to the presence of multiple scattering. This detrimental aspect can be mitigated using cross-correlation schemes with two incident beams and two detectors[40,46]. In the case of 3D cross-correlation, the suppression of multiple scattering comes from the experimental geometry. We use this technique to quantify both the static and dynamic properties of concentrated pNIPAM-pEGd suspensions. Our instrument is manufactured by LS Instruments and enables measurements at scattering angles $\theta \in [20, 140]°$. The detectors are two avalanche photo-diodes and the radiation source is a He−Ne laser that produces polarized light with a vacuum wavelength $\lambda_0 = 632.8$ nm.

**3D dynamic light scattering.** We use 3D dynamic light scattering to measure a relaxation time even in the presence of multiple scattering. In dilute samples, both multiple scattering and inter-particle interactions are normally absent; this simples analyzing the scattering dynamics to obtain the relaxation time associed to particle diffusion, from which, using the Stokes-Einstein relation, we can determine the particle size. At high concentration, and thus in the presence of multiple scattering and inter-particle interactions, the scattered electric field may nevertheless be considered as the sum of independent and similarly distributed components. As a result, it is Gaussian-distributed[37], and a generalized form of the Siegert relation applies[40]:

$$g_I(q, \tau) - 1 = \beta |g_{12}^E(\tau)|^2, \qquad (2)$$

where $g_I(q, \tau) = \langle I_1(q, t) I_2(q, t + \tau) \rangle / I^2$ is the normalized intensity cross-correlation function, with $I_1$ and $I_2$ the scattered intensities in the two detectors, $I = \sqrt{\langle I_1 I_2 \rangle}$ the mean scattered intensity and $\tau$ the correlation time. In addition, $|g_{12}^E(q, \tau)|^2 = \frac{|G_{12}^E(q, \tau)|^2}{\langle I^{(1)}(q) \rangle^2}$, with $I^{(1)}(q)$ the single-scattered intensity and $|G_{12}^E(q, \tau)|^2 = \frac{\epsilon_s}{4\mu_0} \langle E_1^{(1)}(q, t) E_2^{(2)*}(q, t + \tau) \rangle \langle E_1^{(1)*}(q, t) E_2^{(2)}(q, t + \tau) \rangle$ the electric-field cross-correlation function obtained from the scattered electric field $E_i^{(j)}$ in detector $i$ resulting from beam $j$ and its complex conjugate. The quantity $\beta = \beta_0 \frac{\langle I^{(1)}(q) \rangle^2}{I^2}$ is the so-called intercept quantifying the signal-to-noise ratio, with $0 < \beta_0 < 1/4$ the intercept in the dual beam-detector design in the absence of multiple scattering and perfect optical alignment. Using a dilute calibration sample, we obtain $\beta_0$ at every $q$.

Experimentally, we measure $g_I(q, \tau)$ using a correlator, at different $q$. For our pNIPAM-pEGd system, we find that at $\zeta = 1.5$, $|g_{12}^E|^2$ is well described by a stretched exponential, $|g_{12}^E|^2 = \exp[-(\tau/\tau_\alpha)^p]$, which can be regarded as an average over a distribution of simple exponentials, each associated with certain characteristic time. The structural relaxation time at a given $q$ is given by $\tau_\alpha$, and the stretching exponent $p$ is related to the variance in the distribution of exponentials. We note that while for highly concentrated samples with slow dynamics, ergodicity is often lost, requiring sample rotation for performing ensemble averages[68], in our suspensions both at $\zeta = 1.5$ and $\zeta = 3.0$, the samples are ergodic and the dynamics decorrelate at sufficiently long $\tau$.

**3D static light scattering.** We obtain the mean single-scattered intensity from the intercepts $\beta$ and $\beta_0$, and the mean scattered intensity, $\langle I^{(1)}(q) \rangle = I \sqrt{\frac{\beta}{\beta_0}}$. We do this at different $q$, and use the particle

number densities in the dilute and concentrated states, as well as the form factor measurement measured for dilute suspensions, to obtain the measurable structure factor. Even if at the relatively high $\zeta$ we work with, particles might likely have slightly deswollen, this does not have any major influence on our results. In fact, the $q$-position of $S(q)$ is located at $q^* R_h^{eff} \approx \pi$, as expected; see Section 1 of Supplementary Information.

**Correlation function analysis.** For concentrated pNIPAM-pEGd samples, either all of the intensity correlation function or its long-time decay is well described as a stretched exponential. From the Siegert relation, we then have:

$$(g_I(\tau) - 1)/\beta = |g_{12}^E(\tau)|^2 = \exp[-(\tau/\tau_\alpha)^p], \quad (3)$$

which after two logarithmic transformations can be re-written as:

$$\log(-\log[(g_I(\tau) - 1)/\beta]) = p(\log(\tau) - \log(\tau_\alpha)). \quad (4)$$

A linear fit of the data then allows obtaining both $p$ and $\tau_\alpha$. Note how at $\zeta = 3.0$, we can easily observe the short-time and long-time relaxations; see Section 2 of Supplementary Information.

## Simulation details

We perform Langevin dynamics simulations with $N = 5000$ colloid particles of mass $m$ interacting with a MH potential[51] as described in the text. Numerical simulations are performed with the LAMMPS package[69].

The link between elasticity and the structural changes in the pNIPAM-pEGd microgels due to temperature variations is encoded in changes in the extent of the different shells comprising the spherical particles. Table S1 compiles the experimental $R_g/R_h$ values at different $T$, and the values of $\sigma_{eff}$, $\sigma_{core}$, and $\sigma_{mid}$, so that $\sigma_{core}/\sigma_{eff} = R_g/R_h$.

The units of length, energy, and mass in the simulations are $\sigma_{eff}$, $k_B T$, and $m$, respectively. Thus, time is measured in units of $t = \sqrt{m\sigma_{eff}^2/(k_B T)}$. The integration time-step is fixed to $\delta t = 0.002$. Simulations are performed with a friction coefficient $\xi = 10 \, m/t$, and a polydispersity of 10% to approximate experimental conditions; this is achieved by considering 4 different particle sizes. At a different state point $[T, \zeta]$, where $\zeta = NV/V_{sample}$, with $V = \frac{\pi}{6} \langle \sigma_{eff}^3 \rangle$ the particle volume and $V_{sample}$ the volume of the simulation box, we compute the static structure factor $S(q)$ of the polydisperse system,

$$S(q) = \frac{1}{Nb^2(q)} \left\langle \sum_{i,j}^N b_i(q)b_j(q) \exp\left[-i\mathbf{q} \cdot \left(\mathbf{r}_i - \mathbf{r}_j\right)\right] \right\rangle, \quad (5)$$

with $\mathbf{r}_i$ and $b_i(q)$ the position and scattering amplitude of the $i$th particle. Note the latter is different for each particle size, and is computed following ref. 70. The average $b^2(q)$ is obtained by considering all particles, and $\langle \cdots \rangle$ stands for averages over different configurations. We also compute the dynamic structure factor,

$$F_c(q, t) = \frac{1}{S(q)N} \left\langle \sum_{i,j}^N \exp\left[i\mathbf{q} \cdot \left(\mathbf{r}_i(t + t') - \mathbf{r}_j(t')\right)\right] \right\rangle, \quad (6)$$

at $q$ values around the main peak of the static structure factor.

## Data availability

All data generated in this study have been deposited in the Zenodo database.

## Code availability

Numerical codes are available from the corresponding author upon request.

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

## Acknowledgements

We acknowledge the support from MCIN/AEI/10.13039/501100011033/ FEDER, UE (PID2024-156067NB-I00), the 2021 SGR 00450, and the SNSF (200020_184839). We also thank the Spanish University Ministry via the PRE2019-091743 grant associated to project GC2018-097842-B-I00, and the Department of Research and Universities of the Generalitat de Catalunya through Beatriu de Pinós program Grant (Contract No. 2022 BP 00156). We finally gratefully acknowledge the RES resources provided by FinisTerraeIII and MareNostrum 5 at Centro de Supercomputación de Galicia (CESGA) and Barcelona Supercomputing Center (BSC), project FI-2024-2-0033.

## Author contributions

A.A.-G., J.C.-R., P.N.S., J.R.-F., and A.F.-N. contributed to all tasks. They all planned research, performed research, analyzed data, interpreted data and wrote the paper.

## Competing interests

The authors declare no competing interests.
