## [Transparent Peer Review file · Nature Communications]

Structure-Dynamics decoupling in soft-colloid suspensions

Corresponding Author: Dr José Ruiz-Franco

Version 0:

Reviewer comments:

Reviewer #2

(Remarks to the Author)

The manuscript "Structure-Dynamics decoupling in soft-colloid suspensions" by Arenas-Gullo et al reports thought-provoking results about the relation between static structure and structural relaxation in relatively common microgel suspensions.

The authors report that for soft microgels, the structural relaxation occurs on the same time scale for a range of length scales around to the first-neighbor distance, where usually a strong length-scale dependence is observed, a phenomenon usually associated to de Gennes narrowing.

They observe results with a similar trend in molecular dynamics simulations of soft spheres approximating the micro-mechanics of actual microgels.

I found the experimental data quite compelling and I appreciated the effort made to approach the issue simultaneously with numerical simulations. I found the manuscript well written and quite pedagogical. Overall I support this work to be published in some form in Nature Communications, but I believe a few improvements would benefit its impact.

In particular, I felt a need for a more extended discussion regarding the origin of the phenomenon, which I associate to the numerical data being relatively underexploited beyond a mere comparison with the experimental data.

Separately, I also missed a short discussion about the expectation of structure/property coupling. In supercooled liquids and glasses, the relation between structure and slowing down of the dynamics is known to be elusive ("Glasses are structurally undistinguishable from liquids"), the relaxation time increases by orders of magnitude with only minute change in the structure factor. In this specific sense, structure and property are decoupled in a glass. The authors here define a decoupling between structure and property in an other sense, so perhaps it would be interesting to more clearly mark the distinction between their definition of structure/property decoupling and the one I mention here, which I believe is pretty commonly mentioned in the literature.

I also have other points the authors might want to consider.

* In simulations the structure/property decoupling is much less obvious than in experiments. In particular, even for the lowest temperatures in Fig. 3 (and Figs. 8 and 9), there is still a peak in τ_{α} near the first peak of the structure factor, and the value at the peak is much larger than the value in the small- q limit. (This relative height is by the way what is supposed to be plotted in Fig. 3c but I did not understand how $\Delta \tau_{\alpha} = \tau_{\alpha}(q^*)/\tau_{\alpha}(0) - \tau_{\alpha}(0)/\tau_{\alpha}(0)$ can show values of order 1, or even vanish for $U_{mid}=200$ and low T , when the plots in Figs. 3a, 8 and 9 always show prominent peaks.)

Here this relates to my main concern above: it would be nice to further interrogate the numerical data to understand the origin of the attenuation of the peak in $\tau_{\alpha}(q)$, and why it does not show a complete disappearing.

* q -independent relaxation times were already measured in colloidal hard spheres by van Meegen and Underwood [Phys. Rev. E 49, 4206 (1994)]. Arguably this is not a common observation (Ref 18 shows a different result), but it would be interesting to mention it, especially because in van Meegen and Underwood particle are hard.

* I am a bit confused not to see any beta-relaxation in the experimental data. There is a stub of it in the numerical data (especially in Fig. 9, but it is also definitely there in Fig. 3), but it is not visible in the experiments. Given the densities involved here, I would have expected a more clearly visible two-step relaxation. Is it a sign that simulations and experiments do not explore the same region of the phase diagram? (Could it be the reason behind the fact that simulations never observe

complete erasing of the peak in $\tau_\alpha(q)$?)

* For hard colloids, there is an extended literature about mode-coupling predictions exactly centered around the same questions than addressed here, only showing a strong correlation between the q dependences of the structure factor and the relaxation time [e.g Fuchs and Mayr, PRE 60(5) 5742 (1999), Fuchs et al, PRA 45(2), 898 (1992), van Megen and Underwood, PRL 70(18), 2766 (1993), van Megen et al 58(5), 6073 (1998)]. I think this would be useful to say a word about why the authors expect these results not hold here. I understand that the argument given in the manuscript is that particle softness is the root of the decoupling, but it is not obvious why this would be from the point of view of mode-coupling theory.

Reviewer #3

(Remarks to the Author)

This manuscript starts well and raises the reader's interest by suggesting a potential violation of the de Gennes narrowing relation. The authors present high-quality data. However, aspects of the data treatment and interpretation remain unclear.

To begin with, the scaling of the data is not entirely transparent. In this field, the effective volume fraction is typically defined via the swelling ratio, which appears to be known in this study. Why then do the authors not follow this conventional procedure? Why do they not show the raw data for $R_H(T)$ and $R_g(T)$. What exactly is being compared — samples with the same effective volume fraction? If this is the case, the observed change in softness between 17.5 °C and 15 °C is quite dramatic and unexpected.

Next, the authors determine what they refer to as the effective structure factor. However, they do not use a consistent prefactor across all measurements (arbitrary units as usual should replace a.u.). What they show appears to be the measurable structure factor — unless something else is intended. I recommend that the authors adopt this standard terminology if no distinction is being made. It is also important to note that the measurable structure factor can be strongly influenced by polydispersity. It would be helpful to provide information on the polydispersity of the colloidal system, which I could not find in the manuscript.

The central result of the work appears to be the variation of $\tau(q)$ vs. q as the temperature changes from 10 °C to 17.5 °C and beyond. The authors interpret this as a signature of a violation of the de Gennes narrowing. I am not convinced. At 10 °C and 15 °C, the samples show no pronounced peak in $S(q)$; the difference between the high- q values and the putative peak is barely 10%. Indeed, $\tau(q)$ at low q does not decay as expected, but even here $S(q)$ only drops only a little for $T < 17c$. Polydispersity could play a role in this regime, in a way reminiscent of the effect of incoherent scattering in quasielastic neutron scattering, where the signal is sensitive to self-diffusion.

Therefore, my main criticism is that the authors do not offer a convincing explanation for the dramatic change in $S(q)$ between 17.5 °C and 10 °C. Is this related to the observed drop in R_g/R_h from 17.5 °C to 15 °C, and has this been reported previously in the literature?

I have not studied the numerical modelling results in detail, as I believe this paper's conclusions hinge on the experimental results, which are not sufficiently convincing in their current form. Finally, in my view, the discussion and conclusions need to be expanded to distinguish the essential claims from the supporting evidence clearly. This would help the reader better understand the core message of the paper and the basis on which it rests.

Version 1:

Reviewer comments:

Reviewer #2

(Remarks to the Author)

The revision of the manuscript substantially improved the discussion of the experimental results with respect to the numerical results in this work and other results in the literature. The fact that results are unexpected and thought-provoking remains, and I think this work will stimulate others in trying to investigate similar effects in experiments of soft particles, trying to numerically and theoretically model the deformation of particles and how it affects the structural relaxation.

I recommend for publication.

Reviewer #3

(Remarks to the Author)

The authors have improved the manuscript and have taken into account some of my suggestions. However, some of the wording—particularly in the abstract and introduction—is still not well written. For example, the first two sentences of the abstract: what do the authors even mean? They should stick more closely to their actual claims and remove vague language such as “a paradigm in materials science is that the internal structure of a material determines its macroscopic properties.” What structure? What properties?

The fact that the observation of interest (the loss of structure in $\tau(q)$) coincides with the sharp drop in the peak of $S(q)$

(17.5°C and below) is a matter of concern for me. If $S(q)$ remained unchanged while $\tau(q)$ lost structure, the observation would be clear-cut—but that is not the case. As a result, I do not know what's happening, and the authors have not convinced me.

The system is at a generalized (or effective) volume fraction of 1.5—it should already be a glass or jammed. I do not understand why it is relaxing on such short time scales.

I spent quite some time studying the revised manuscript, but it remains difficult to access. The parameter space is very complex, with many renormalized units. I cannot invest the time required to make a clear recommendation. At the same time, I do not want to issue a firm negative recommendation based on an incomplete analysis.

Reviewer #4

(Remarks to the Author)

This manuscript investigates the relationship between particle stiffness, the cage effect, and fragility in colloidal and soft particle systems. Two central findings emerged. First, softer particles exhibit weaker cage effects and behave as strong liquids when analyzed using the actual volume fraction. This weakening of the cage effect is consistent with a reduction in de Gennes narrowing. This indicates diminished coupling between structural correlations and particle dynamics. This finding is also consistent with prior studies (e.g., Refs. 54 and related works). Therefore, it is a physically reasonable interpretation. Second, when the effective volume fraction is calculated using Voronoi volumes, the dependence of fragility on particle softness disappears, resulting in a striking universal scaling law. This finding goes beyond the insights of Ref. 54, offering a more general and unifying description of fragility behavior across systems with varying particle stiffness.

The authors responded sincerely and constructively to previous reviewers' questions, leading to substantial improvements in the manuscript. The clarifications, particularly those regarding the measurement and interpretation of the effective volume fraction and the robustness of the observed universal behavior, have significantly enhanced the transparency and scientific rigor of the study.

However, I believe it is crucial for the authors to address the relationship between the amplitude of the first peak of $S(q)$ and the corresponding variation of $\tau(q)$ more quantitatively, especially within the range of 10-15°. While the peak is visible, as Reviewer 3 noted, the current presentation does not clarify whether the ~10% change in $S(q)$ sufficiently explains or contrasts with the pronounced changes in $\tau(q)$. Providing such an explicit comparison is necessary, in my view, to substantiate the claim of a breakdown (or weakening) of de Gennes narrowing. Without this additional analysis, the interpretation is less convincing.

Overall, I find the manuscript highly original, scientifically sound, and of substantial interest to the community. I believe it merits publication in Nature Communications once the above issue is addressed.

Version 2:

Reviewer comments:

Reviewer #4

(Remarks to the Author)

The authors have responded sincerely to reviewer 3, who expressed disagreements, by thoroughly comparing their work with relevant prior studies. I believe their responses are also scientifically very sound on the validity of experimental results and the simulations. Considering the above, as stated in the previous review, I think the results of this paper are highly significant and worthy of publication in Nature Communications.

In the following we provide detailed answers to the reviewers' comments and questions and relevant changes included in the paper. To latter are reported in blue. We also provide a track-change version of the manuscript for reference.

REPLY TO THE REPORT OF REVIEWER 2

The manuscript "Structure-Dynamics decoupling in soft-colloid suspensions" by Arenas-Gullo et al reports thought-provoking results about the relation between static structure and structural relaxation in relatively common microgel suspensions.

The authors report that for soft microgels, the structural relaxation occurs on the same time scale for a range of length scales around to the first-neighbor distance, where usually a strong length-scale dependence is observed, a phenomenon usually associated to de Gennes narrowing. They observe results with a similar trend in molecular dynamics simulations of soft spheres approximating the micro-mechanics of actual microgels.

I found the experimental data quite compelling and I appreciated the effort made to approach the issue simultaneously with numerical simulations. I found the manuscript well written and quite pedagogical. Overall I support this work to be published in some form in Nature Communications, but I believe a few improvements would benefit its impact.

We thank Reviewer 2 for the appreciation of our work.

In particular, I felt a need for a more extended discussion regarding the origin of the phenomenon, which I associate to the numerical data being relatively underexploited beyond a mere comparison with the experimental data.

In addition to the information we have added to the manuscript in connection to the referees' comments, we have also extended the discussion of our results to more extensively discuss the origin of the decoupling between static and dynamic properties. Below, we provide a summary of the main additions included in the revised version of the manuscript:

- Expanded discussion on how mode-coupling theory may fail to capture the dynamic glass transition due to internal degrees of freedom associated with particle deformation.
 - Strengthened the connection between simulations and experiments.
-

- Clarified the structural particle changes associated with the transition in particle softness.
- Emphasized the difference between de Gennes narrowing and the structure/dynamics "decoupling" on the approach to the glass
- Provided additional experimental details.
- Added relevant references

Separately, I also missed a short discussion about the expectation of structure/property coupling. In supercooled liquids and glasses, the relation between structure and slowing down of the dynamics is known to be elusive ("Glasses are structurally undistinguishable from liquids"), the relaxation time increases by orders of magnitude with only minute change in the structure factor. In this specific sense, structure and property are decoupled in a glass. The authors here define a decoupling between structure and property in an other sense, so perhaps it would be interesting to more clearly mark the distinction between their definition of structure/property decoupling and the one I mention here, which I believe is pretty commonly mentioned in the literature.

As Reviewer 2 correctly points out, in supercooled liquids and glasses, the structure factor exhibits only minor changes while the relaxation time increases dramatically. We completely agree in that this could be seen as a structural/dynamic decoupling, but in a sense that is different from ours, and ultimately from that associated to de Gennes narrowing. We also agree with Reviewer 2 in that discussing the situation of liquids on the approach to the glass and the different type of structure/dynamics decoupling will enhance the breadth of our work. We have therefore added a discussion of this point in the revised version of the paper.

In supercooled liquids and glasses, the onset of slow dynamics is governed by the increase in the structural relaxation time τ_α as the volume fraction ϕ increases, while the static structure factor $S(q)$ remains essentially unchanged [Bengtzelius, U. et al., Journal of Physics C: solid state Physics 17, 5915 (1984), Di Cola, E. et al., J. Chem. Phys 131, 14 (2009)]. This scenario reflects a structure-dynamics decoupling, albeit in a sense that differs from the one associated with de Gennes narrowing. The latter occurs at constant ϕ and reflects how the dynamics slows down at length scales comparable to the interparticle separation. Hence, τ_α increases around the wave vector q at which $S(q)$ is peaked [Binder, K, Kob, W.: Glassy Materials and Disordered Solids: An Introduction to Their Statistical Mechanics (2011)]. While the coupling between structure and dynamics holds for many soft materials, ranging from hard-spheres [Fuchs, M. et al., Physical Review A 45, 898 (1992), Fuchs, M. et al., Physical Review E 60, 5742 (1999)] to emulsion and polymeric systems [Gang, H. et al., Physical Review E 59, 715 (1999), Colmenero, J. et al., J. Chem. Phys 139, 4 (2013)], we demonstrate here that particle softness can introduce additional relaxation mechanisms to allow the system to relax on shorter time scales. As a result, even in regimes where structural features are prominent, these do not alone govern the dynamic properties, and hence, τ_α can become q -independent.

In simulations the structure/property decoupling is much less obvious than in experiments. In particular, even for the lowest temperatures in Fig. 3 (and Figs. 8 and 9), there is still a peak in τ_α near the first peak of the structure factor; and the value at the peak is much larger than the value in the small- q limit. (This relative height is by the way what is supposed to be plotted in Fig. 3c but I did not understand how $\Delta\tau_\alpha = \tau_\alpha(q^)/\tau_0(q^*) - \tau_\alpha(0)/\tau_0(0)$ can show values of order 1, or even vanish for $U_{mid} = 200$ and low T , when the plots in Figs. 3a, 8 and 9 always show prominent peaks.) Here this relates to my main concern above: it would be nice to further interrogate the numerical data to understand the origin of the attenuation of the peak in $\tau_\alpha(q)$, and why it does not show a complete disappearing.*

In Fig. 3c, as Reviewer 2 correctly notes, we plot the relative height $\Delta\tau_\alpha$ versus T . However, we would like to emphasize that $\Delta\tau_\alpha$ does not vanish for $U_{mid} = 200k_B T$ at low temperature. There is a factor 10^3 indicated in the y -axis that might have been hard to see. We have replotted the corresponding graph in Fig. 3c to improve the visualization of the results.

In Fig. R1, we show the normalized structural relaxation time as a function of the intermediate shell elasticity U_{mid} at the lowest temperature studied, $T = 10^\circ\text{C}$. The relative height of the peak decreases with U_{mid} , but it does not disappear, as Reviewer 2 indicates. We rationalize this with the fact we are simulating spherical particles with a fixed size σ_{eff} . At the high ζ values we explore, these particles overlap one with another, intrinsically imposing an energy barrier that must be overcome for structural relaxation to occur. Thus, the peak observed in $\tau_\alpha(q)$, even at low T and at q^* values corresponding to the main peak of $S(q)$, is related to how the overlaps imposed by the positional correlations affect the subsequent dynamics. This behavior contrasts with our experimental system, where the particles have internal degrees of freedom at play when there is deformation. This deformability allows the pNIPAM-pEGd microgels to adjust their shape to fill available space without any significant overlaps, thereby lacking the energetic cost present in simulation in relation to the structural relaxation. The key role played by particle softness and internal degrees of freedom in the experiment, effectively encapsulated in the simulations through a Multi-Hertzian potential and inter-particle overlaps, then results in the difference with respect to complete or partial erasure of the $\tau_\alpha(q)$ peak. The simulations, nevertheless, clearly show that increasing the effective single-particle softness decreases the relevance of the dynamic peak relative to the structural structure factor peak.

Figure R1: Normalized structural relaxation time $\tau_\alpha(q^*)/\tau_0(q^*)$ as a function of the intermediate shell elasticity U_{mid} at $T = 10^\circ\text{C}$.

We now discuss this in our manuscript through the following text:

Thus, our simulation results clearly show that single-particle elasticity can significantly affect the coupling between structure and dynamics. However, unlike the experimental observations, a complete erasure of the peak in τ_α is not observed in simulations. In our model, particle deformation is captured through particle overlaps, which impose an energy barrier that must be overcome to allow for structural relaxation. Experimentally, our pNIPAM-pEGd particles can deform to accommodate crowding without any meaningful overlaps, suppressing these energy barriers. As a result, the peak in $\tau_\alpha(q)$ observed in simulations at low T is absent in experiments, highlighting the distinct way particle softness manifests in the two approaches.

q-independent relaxation times were already measured in colloidal hard spheres by van Meegen and Underwood [Phys. Rev. E 49, 4206 (1994)]. Arguably this is not a common observation (Ref 18 shows a different result), but it would be interesting to mention it, especially because in van Meegen and Underwood particle are hard.

We thank Reviewer 2 for bringing this reference to our attention. As Reviewer 2 mentions, the observation of a constant τ_α vs q is not common. In the paper by van Meegen and Underwood referred to by Reviewer 2, the authors report that the structural relaxation time shows no systematic wave-vector dependence. This is in contrast to experimental observations with the same system by Segré and Pusey [PRL 77(4), 771 (1996)]. Additional studies on hard spheres by Fuchs *et al.* [PRA 45(2), 898 (1992)], and recent work

with metallic glasses by B. Ruta et al [PRL 125(5), 055701 (2020), Comm. Physics 5:316 (2022)] also indicate that the structural relaxation time is q dependent.

This apparent discrepancy might arise from fitting artifacts or limitations in the data analysis in the van Megen and Underwood study, which could have masked any possible q dependence. Indeed, we have performed a simple analysis using the data published by van Megen and Underwood [Phys. Rev. E 49, 4206 (1994)]. In this analysis, we estimate τ_α as the time at which the collective scattering function decays to $1/e$. We find that the τ_α estimated in this way depends on q . Given the majority of experimental and theoretical work, including those kindly recommended by Reviewer 2, show a marked q -dependence of the structural relaxation time, we have chosen not to include the van Megen and Underwood paper in our citations, focusing instead on the references that better align with the current understanding in the field. We hope the referee finds this reasonable.

I am a bit confused not to see any β -relaxation in the experimental data. There is a stub of it in the numerical data (especially in Fig. 9, but it is also definitely there in Fig. 3), but it is not visible in the experiments. Given the densities involved here, I would have expected a more clearly visible two-step relaxation. Is there a sign that simulations and experiments do not explore the same region of the phase diagram? (Could it be the reason behind the fact that simulations never observe complete erasing of the peak in $\tau_\alpha(q)$?)

As Reviewer 2 correctly points out, pNIPAM-pEGd suspensions at $\zeta = 1.5$ do not exhibit any β -relaxation mechanism at any temperature. This relaxation process is typically associated with the microscopic motions of particles that are temporarily trapped by their neighbors, a phenomenon known as caging. Note, however, that for $T > 17.5^\circ\text{C}$, the stretching exponent p is roughly 0.75, corresponding to a stretched exponential behavior and highlighting that the microscopic dynamics of the suspension are not merely diffusive. In addition, at $\zeta = 3.0$, β -relaxation becomes evident for $T \geq 22^\circ\text{C}$. This observation suggests that particle softness facilitates cage escape by allowing particles to deform and rearrange more easily, which can, in turn, lead to faster microscopic relaxation and a reduction of caging effects.

We have emphasized this observation in our manuscript:

Typically, in supercooled liquids and glasses, $|g_{12}^E|^2$ exhibits two decays [W. Gotze: Complex Dynamics of Glass-forming Liquids: A Mode-coupling (2009); W. Van Megen, *et al.*, PRL 70, 2766 (1993), M. Laurati, *et al.*, PCCP 20, 18630–18638 (2018)]. The first decay, present at short times, is known as the β -relaxation mechanism and is associated with the microscopic motion of particles within their instantaneous nearest-neighbor cages. The second decay, emerging at longer times, is attributed to the α -relaxation mechanism, corresponding to the structural relaxation of the system. In our case, we observe that $|g_{12}^E|^2$ exhibits a single decay, described by a stretched exponential, as shown by the corresponding fits in Fig. 2(b).

In addition, we have added the following sentence:

For $\zeta = 3.0$, $|g_{12}^E|^2$ exhibits a two decay behavior depending on T , indicating that caging effects become relevant at this higher ζ .

The absence of β -relaxation at $\zeta = 1.5$ in experiments and its presence in simulation could contribute to the $\tau_\alpha(q)$ peak seen in the latter even at low T , in connection to the already highlighted differences in both approaches.

For hard colloids, there is an extended literature about mode-coupling predictions exactly centered around the same questions than addressed here, only showing a strong correlation between the q dependences of the structure factor and the relaxation time [e.g Fuchs and Mayr, PRE 60(5) 5742 (1999), Fuchs et al, PRA 45(2), 898 (1992), van Megen and Underwood, PRL 70(18), 2766 (1993), van Megen et al 58(5), 6073 (1998)]. I think this would be useful to say a word about why the authors expect these results not hold here. I understand that

the argument given in the manuscript is that particle softness is the root of the decoupling, but it is not obvious why this would be from the point of view of mode-coupling theory.

We thank Reviewer 2 for bringing these references to our attention, which we now cite in our paper.

Our results indicate that $S(q)$ alone is not sufficient to predict the dynamics. It is necessary to consider the role of the particles' internal degrees of freedom, as these can facilitate relaxation processes that are not included in mode-coupling theory. We now mention this in the revised manuscript:

In this direction, we note that MCT relies on $S(q)$ to predict the wave vector-dependent dynamics and the growth of τ_α as the system approaches the glass [W. Gotze: *Complex Dynamics of Glass-forming Liquids: A Mode-coupling* (2009), L. M. Janssen, *Frontiers in Physics* 6, 97 (2018)]. However, in the case of soft particles, like the pNIPAM-pEGd microgels studied here, additional relaxation pathways arise from the internal degrees of freedom associated with particle deformability and softness, enabling dissipative mechanisms that are not captured by $S(q)$ alone. As a result, the assumption that structural correlations fully determine the dynamics breaks down. This suggests that MCT, in its standard form, may not account for the relaxation mechanisms present in ultrasoft systems, and that modifications to the theory may be required to incorporate the influence of particle softness.

Once again, we very much thank Reviewer 2 for the constructive comments on our work. They have greatly helped us revise and improve our manuscript.

REPLY TO THE REPORT OF REVIEWER 3

This manuscript starts well and raises the reader's interest by suggesting a potential violation of the de Gennes narrowing relation. The authors present high-quality data. However, aspects of the data treatment and interpretation remain unclear.

To begin with, the scaling of the data is not entirely transparent. In this field, the effective volume fraction is typically defined via the swelling ratio, which appears to be known in this study. Why then do the authors not follow this conventional procedure? Why do they not show the raw data for $R_H(T)$ and $R_g(T)$? What exactly is being compared — samples with the same effective volume fraction? If this is the case, the observed change in softness between 17.5°C and 15°C is quite dramatic and unexpected.

We are thankful to Reviewer 3 for the various comments, which we progressively address below.

What we call generalized volume fraction is defined as $\zeta = nV_0$, with n the number density and V_0 the volume of the particle, $V_0 = \frac{4}{3}\pi R_h^3$, in dilute conditions. Note R_h , and thus V_0 , can correspond to both swollen and deswollen states. In the literature, ζ is sometimes also referred to as effective volume fraction. However, since ζ can be larger than 1, we prefer to call it generalized volume fraction, as values larger than 1 do not effectively correspond to physically meaningful values of the real volume fraction ϕ .

We did not include $R_h(T)$ and $R_g(T)$ in the main text, since both quantities were extensively studied and discussed previously in [J. Clara-Rahola *et al.*, JCP 136, 21, (2012)]. We feel that the ratio of the two quantities, R_g/R_h , is the most relevant quantity, as it reflects deviations from hard-sphere behavior and enables comparisons with values for soft particles. For this reason, we present this ratio in Fig. 1a.

The data at $\zeta = 1.5$ indeed shows significant differences from 17.5°C to 15°C , as reported in Fig. 2a. While the corresponding variation of R_g/R_h might appear modest, it is significant when contrasted with the behavior of pure PNIPAM microgels, where R_g/R_h remains essentially unchanged in this temperature range. The shift in softness we observe reflects intrinsic modifications in the single-particle properties, as previously reported through spectroscopy and dynamic light scattering measurements on PNIPAM-PEG block copolymers [R. Motokawa *et al.*, Macromolecules 38, 5748–5760 (2005); J. Clara-Rahola *et al.*, J. Chem. Phys. 136, 21 (2012)]. These studies indicated that hydrophobicity begins to emerge around $T \approx 17^\circ\text{C}$, indicating that water behaves as a selective solvent for the PEG blocks. This results in PEG chains remaining more swollen than the PNIPAM segments, inducing internal microphase separation and heterogeneities within the microgel network, ultimately leading to modifications in the particle softness.

We have included these aspects in our manuscript by adding the following text:

The less usual transition from a close-to-star-polymer to a soft sphere structure at lower temperatures has been attributed to the presence of pEGd [J. Clara-Rahola *et al.*, J. Chem. Phys. 136, 21 (2012)]. At $T \approx 17^\circ\text{C}$ pEGd segregates from pNIPAM, inducing internal microphase separation and heterogeneities within the microgel network [R. Motokawa *et al.*, Macromolecules 38, 5748–5760 (2005)]. This segregation causes the shell of the soft sphere to further stretch out at low T , bringing about a form factor that is well described using star-polymer models [J. Clara-Rahola *et al.*, J. Chem. Phys. 136, 21 (2012)], and consequently modifying the particle softness. The morphological change of the particle with T is schematically sketched on top of Fig. 1(a).

Next, the authors determine what they refer to as the effective structure factor. However, they do not use a

consistent prefactor across all measurements (arbitrary units as usual should replace a.u.). What they show appears to be the measurable structure factor — unless something else is intended. I recommend that the authors adopt this standard terminology if no distinction is being made. It is also important to note that the measurable structure factor can be strongly influenced by polydispersity. It would be helpful to provide information on the polydispersity of the colloidal system, which I could not find in the manuscript.

We have followed the Reviewer suggestion and replaced a.u. with arb. units in all relevant plots. Additionally, we have clarified in the revised manuscript that the structure factor shown is not scaled by any prefactor and that therefore corresponds to the standard measurable structure factor. We have also removed the term effective structure factor to avoid confusion and now refer to it using the standard terminology.

The polydispersity in our colloidal system is in the range between 5% – 10% [J. Clara-Rahola et al., J. Chem. Phys. 136, 21 (2012)]. We now include this information in the revised manuscript. Previous work with pNIPAM microgel suspensions with a polydispersity of approximately 30% [B. Zhou et al., PRE 108, 054604 (2023)], has shown that the main effect of polydispersity is to suppress the characteristic oscillations of the liquid-like structure factor and to increase the magnitude of $S(q)$ at $q \rightarrow 0$, while only slightly reducing the height of the main peak. Based on these observations, we expect polydispersity will not significantly affect the static/dynamic decoupling observed experimentally.

The central result of the work appears to be the variation of $\tau(q)$ vs. q as the temperature changes from 10°C to 17.5°C and beyond. The authors interpret this as a signature of a violation of the de Gennes narrowing. I am not convinced. At 10°C and 15°C , the samples show no pronounced peak in $S(q)$; the difference between the high- q values and the putative peak is barely 10%. Indeed, $\tau(q)$ at low q does not decay as expected, but even here $S(q)$ only drops only a little for $T < 17^\circ\text{C}$. Polydispersity could play a role in this regime, in a way reminiscent of the effect of incoherent scattering in quasielastic neutron scattering, where the signal is sensitive to self-diffusion. Therefore, my main criticism is that the authors do not offer a convincing explanation for the dramatic change in $S(q)$ between 17.5°C and 10°C . Is this related to the observed drop in R_g/R_h from 17.5°C to 15°C , and has this been reported previously in the literature?

Indeed, R_g/R_h drops from 17.5°C to 15°C . This drop is directly related with significant changes in the form factor $P(q)$. While $P(q)$ at high T is well described using a soft-sphere model, at low T it is described with a star-polymer model [J. Clara-Rahola et al., J. Chem. Phys. 136, 21 (2012)]. The measurable structure factors we report are normalized using the scattering intensity $I(q)$ obtained under dilute conditions. However, at $\zeta = 1.5$ or $\zeta = 3$, in addition to shrinking, which we do account for, changes in internal structure may occur and these may occur differently depending on T , since the different internal particle-structure might respond differently to concentration. In addition, polydispersity at high ζ may differ from that in dilute conditions, depending on particle morphology. All these effects could influence $S(q)$ and are not all taken into account in our normalization. Nevertheless, there is a clear increase from low q to the main peak at all T , and a visible, albeit milder, peak in $S(q)$, at the lower T , all the while $\tau_\alpha(q)$ remains essentially flat. Additionally, as shown in Fig. 5(a) of Extended Data 1, the $S(q)$ peak shifts in a consistent way throughout the whole temperature range we explore.

These aspects are all now mentioned in the revised version of the manuscript:

In addition, polydispersity at high ζ may differ from that in dilute conditions, and even at different T , due to changes in the particle morphology. This last aspect could also imply changes in $P(q)$ at $\zeta = 1.5$ relative to the dilute case that are hard to take into account, and that could affect our estimated $S(q)$. As a result, $S(q)$ should be taken as a reasonable estimate of the measurable structure factor.

The less pronounced peak at $T < 17.5^\circ\text{C}$ could relate to changes in $P(q)$ at $\zeta = 1.5$ relative to the dilute case not accounted for in our normalization of $\langle I^{(1)}(q) \rangle$, as mentioned above. The measurable structure factor,

nevertheless, shows a clear increase from low q to the peak, which additionally follows the expected trend when plotted against the particle size R_h^{eff} in Extended Data 1.

I have not studied the numerical modelling results in detail, as I believe this paper's conclusions hinge on the experimental results, which are not sufficiently convincing in their current form. Finally, in my view, the discussion and conclusions need to be expanded to distinguish the essential claims from the supporting evidence clearly. This would help the reader better understand the core message of the paper and the basis on which it rests.

In the revised version of the manuscript, we have expanded the discussion and conclusions to clearly distinguish the essential claims from the supporting evidence.

We are very thankful to Reviewer 3 for the constructive comments, which we believe have helped us to improve our manuscript.

In the following we provide detailed answers to the reviewers' comments and concerns.

REPLY TO THE SECOND REPORT OF REVIEWER 2

The revision of the manuscript substantially improved the discussion of the experimental results with respect to the numerical results in this work and other results in the literature. The fact that results are unexpected and thought-provoking remains, and I think this work will stimulate others in trying to investigate similar effects in experiments of soft particles, trying to numerically and theoretically model the deformation of particles and how it affects the structural relaxation.

I recommend for publication.

We thank Reviewer 2 for the appreciation of our work and for recommending its publication.

REPLY TO THE REPORT OF REVIEWER 3

The authors have improved the manuscript and have taken into account some of my suggestions. However, some of the wording—particularly in the abstract and introduction—is still not well written. For example, the first two sentences of the abstract: what do the authors even mean? They should stick more closely to their actual claims and remove vague language such as “a paradigm in materials science is that the internal structure of a material determines its macroscopic properties.” What structure? What properties?

We would like to start by emphasizing that these parts of the paper did not change with respect to our original submission.

We also respectfully disagree with the observation of the referee. That structure determines properties is indeed the paradigm in material science. There are textbooks at the undergraduate level clearly stating what we assess. Consider the book by Callister, *Material Science and Engineering - An Introduction*. We refer to the 10th edition, published by Wiley, but what we say also appeared in prior editions. In pages 4 and 5, the authors write: *”This interrelationship among processing, structure, properties, and performance of materials may be depicted in linear fashion as in the schematic illustration shown in Figure 1.2. The model represented by this diagram has been called by some the central paradigm of materials science and engineering or sometimes just the materials paradigm. (The term “paradigm” means a model or set of ideas.) This paradigm, formulated in the 1990s is, in essence, the core of the discipline of materials science and engineering. It describes the protocol for selecting and designing materials for specific and well-defined applications, and has had a profound influence on the field of materials ... Experience shows that the properties and phenomena associated with a material are intimately related to its composition and structure at all levels, including which atoms are present and how the atoms are arranged in the material, and that this structure is the result of synthesis and processing”*. Later in the textbook, the authors explicitly mention that this paradigm applies to thermal, electrical, magnetic, optical and mechanical properties. We thus believe that our statements are in no way vague but that they simply reflect general knowledge in materials science. We nevertheless now cite this textbook in the new version of the manuscript in connection with our statement.

We also emphasize that we provide several examples in the paper to illustrate the above-mentioned paradigm, making the statement accessible to the general reader.

The fact that the observation of interest (the loss of structure in $\tau(q)$) coincides with the sharp drop in the peak of $S(q)$ (17.5°C and below) is a matter of concern for me. If $S(q)$ remained unchanged while $\tau(q)$ lost structure, the observation would be clear-cut—but that is not the case. As a result, I do not know what’s happening, and the authors have not convinced me.

As discussed in the manuscript and explained in the previous reply, there is clear evidence that $S(q)$ increases from low q up to the main peak, while the relaxation time remains completely flat. In addition, the position of the peak at low T follows the exact trend found at higher T ; see Fig. 5(a) in Section 1 of the Supplementary Information (previously referred to as Extended Data 1). These two aspects are unambiguous evidence that there is a structural peak at low T .

We also explained in the previous reply and in the manuscript, that there are many factors that influence the detailed shape of $S(q)$. All normalization factors are impossible to access in our experiments, including the exact internal-particle structure and the polydispersity at the ζ of our experiments. It is thus reasonable that the detailed shape of $S(q)$ at low and high T , corresponding to changes in R_{gh} , is not exactly the same and that the height of

the peak relative to the high- q minimum may exhibit mild differences. We believe that these differences nevertheless do not at all detract us from concluding what we state in our paper.

The system is at a generalized (or effective) volume fraction of 1.5—it should already be a glass or jammed. I do not understand why it is relaxing on such short time scales.

We respectfully disagree with the referee. Glassy and jammed states emerge as ζ increases, when particles become confined in microscopic cages formed by their nearest neighbors. These cages are responsible for the slow dynamics. However, because microgels are soft, they can respond to crowding by deswelling and deforming. As a result, microgels can adapt their shape and volume at high concentrations, enabling them to relax on shorter timescales compared to conventional colloidal suspensions, and remain liquid-like for very high ζ . The faster relaxation of these suspensions implies that their viscosity increases with ζ (or concentration) much more slowly than in conventional colloidal suspensions; see, for example, A. N. St. John *et al.*, *J. Phys. Chem. B* 111, 7796–7801 (2007); D. van den Ende *et al.*, *Phys. Rev. E* 81, 011404 (2010); B. Sierra-Martín *et al.*, *Soft Matter* 8, 4141 (2012); M. Pelaez-Fernandez *et al.*, *Phys. Rev. Lett.* 114, 098303 (2015); C. Pellet *et al.*, *Soft Matter* 12, 3710–3720 (2016); A. Scotti *et al.*, *Soft Matter* 16, 668–678 (2020); N. A. Burger *et al.*, arXiv:2508.04244 (2025). Consequently, the glassy and jammed states mentioned by Reviewer 3 occur only at significantly larger ζ values; see also A. Ikeda *et al.*, *Soft Matter* 9, 7669–7683 (2013); D. Vlassopoulos and M. Cloitre, *Curr. Opin. Colloid Interface Sci.* 19, 561–574 (2024).

We also emphasize that despite the references cited above focus on colloidal microgels, similar observations, namely the manifestation liquid-like behavior characterized by short-time at $\zeta \gg 1$, has also been reported in other soft-particle suspensions comprised by star polymers, micelles and emulsion droplets.

I spent quite some time studying the revised manuscript, but it remains difficult to access. The parameter space is very complex, with many renormalized units. I cannot invest the time required to make a clear recommendation. At the same time, I do not want to issue a firm negative recommendation based on an incomplete analysis.

The numerical simulations implemented in our work are based on prior experimental and simulation work on colloidal microgels [M. J. Bergman *et al.*, *Nat. Commun.* 9, 5039 (2018)]. Furthermore, similar approaches are commonly employed in the study of soft colloidal particles with an internal polymer architecture, given these type of particles cannot be accurately modeled by a single simple potential, and a full description of all their architectural details is not feasible. We thus use a sound strategy based on a coarse-grained model consisting in overlapping spherically symmetric effective potentials, where the parameters are tuned to reproduce the experimental observations. We emphasize that this approach has become the standard in the literature, as demonstrated in studies of microgel suspensions; see, for example, J. Ruiz-Franco *et al.*, *Soft Matter* 19, 3614–3628 (2023); M. M. Schmidt *et al.*, *Phys. Rev. Lett.* 131, 258202 (2023); J. Ruiz-Franco *et al.*, *Adv. Mater. Interfaces*, 2500242 (2025); T. Hofken *et al.*, *Rheol. Acta*, 1–12 (2025). Similar approaches have also been applied to other colloidal particles with internal polymeric structures, including star polymers [C. N. Likos *et al.*, *Phys. Rev. Lett.* 80, 4450 (1998); M. Watzlawek *et al.*, *Phys. Rev. Lett.* 82, 5289 (1999)], grafted nanoparticles [C. N. Likos *et al.*, *Langmuir* 16, 4100 (2000); D. Parisi *et al.*, *Phys. Fluids* 32, 127101 (2020)], and core-shell particles [P. Kumar *et al.*, *Phys. Rev. E* 72, 021501 (2005); M. A. Sandoval-Puentes *et al.*, *J. Phys.: Condens. Matter* 34, 164001 (2022), J. Martín-Roca *et al.*, *J. Chem. Phys.* 156, 164502 (2022)].

The units used in our numerical simulations are the standard ones commonly adopted in

Molecular Dynamics (MD) simulations [D. Frenkel and B. Smit, *Understanding Molecular Simulation: From Algorithms to Applications*, Elsevier (2023); M. P. Allen and D. J. Tildesley, *Computer Simulation of Liquids*, Oxford University Press (2017); D. C. Rapaport, *The Art of Molecular Dynamics Simulation*, Cambridge University Press (2004)]. A widely used choice is to define all particles with the same mass, $m_i = 1$. As a consequence, particle momenta and velocities become numerically identical, as do forces and accelerations. The unit of length is typically set by choosing a fixed reference length, most often the particle diameter σ . Finally, for repulsive systems, the unit of energy is set by the thermal energy $k_B T$. In this way, the unit of time is defined as $\sqrt{m\sigma^2/k_B T}$. We have therefore followed standard normalization units in our simulations. This is explicitly stated in our manuscript.

We trust that the additional information provided above demonstrates the robustness of our work, additionally clarifying the points raised by Reviewer 3 in this second report.

REPLY TO THE REPORT OF REVIEWER 4

This manuscript investigates the relationship between particle stiffness, cage effect, and fragility in colloidal and soft particle systems, revealing two central findings. (1) Softer particles exhibit weaker cage effects and behave as strong liquids when analyzed using the actual volume fraction. This weakening of the cage effect is consistent with a reduction in de Gennes narrowing, indicating a diminished coupling between structural correlations and particle dynamics, and is also consistent with prior studies (e.g., Refs. 54 and related works), making it a physically reasonable interpretation. (2) When the effective volume fraction calculated from Voronoi volumes is employed, the dependence of fragility on particle softness vanishes, yielding a striking universal scaling law. This finding goes beyond the insights of Ref. 54, offering a more general and unifying description of fragility behavior across systems with varying particle stiffness.

The authors have responded to previous reviewers' questions in a sincere and constructive manner, leading to substantial improvements in the manuscript. The clarifications—particularly regarding the measurement and interpretation of the effective volume fraction, as well as the robustness of the observed universal behavior—have significantly enhanced both the transparency and the scientific rigor of the study.

Overall, I find the manuscript highly original, scientifically sound, and of substantial interest to the community, and I believe it fully merits publication in Nature Communications.

We thank Reviewer 4 for the careful evaluation of our work, for acknowledging our efforts to address the questions and comments raised by the referees in the first review round, and for recommending the manuscript for publication in Nature Communications.